# Differential Transcriptional Programs Reveal Modular Network Rearrangements Associated with Late-Onset Alzheimer’s Disease

**DOI:** 10.3390/ijms26052361

**Published:** 2025-03-06

**Authors:** Alejandra Paulina Pérez-González, Guillermo de Anda-Jáuregui, Enrique Hernández-Lemus

**Affiliations:** 1División de Genómica Computacional, Instituto Nacional de Medicina Genómica, Mexico City 14610, Mexico; agonzalez@inmegen.edu.mx; 2Programa de Doctorado en Ciencias Biomédicas, Unidad de Posgrado Edificio B Primer Piso, Ciudad Universitaria, Mexico City 04510, Mexico; 3Facultad de Estudios Superiores Iztacala, Universidad Nacional Autónoma de México, Mexico City 54090, Mexico; 4Centro de Ciencias de la Complejidad, Universidad Nacional Autónoma de México, Mexico City 04510, Mexico; 5Investigadores por M’exico, Conahcyt, Mexico City 03940, Mexico

**Keywords:** Alzheimer’s disease, LOAD, gene co-expression networks, network analytics

## Abstract

Alzheimer’s disease (AD) is a complex, genetically heterogeneous disorder. The diverse phenotypes associated with AD result from interactions between genetic and environmental factors, influencing multiple biological pathways throughout disease progression. Network-based approaches offer a way to assess phenotype-specific states. In this study, we calculated key network metrics to characterize the network transcriptional structure and organization in LOAD, focusing on genes and pathways implicated in AD pathology within the dorsolateral prefrontal cortex (DLPFC). Our findings revealed disease-specific coexpression markers associated with diverse metabolic functions. Additionally, significant differences were observed at both the mesoscopic and local levels between AD and control networks, along with a restructuring of gene coexpression and biological functions into distinct transcriptional modules. These results show the molecular reorganization of the transcriptional program occurring in LOAD, highlighting specific adaptations that may contribute to or result from cellular responses to pathological stressors. Our findings may support the development of a unified model for the causal mechanisms of AD, suggesting that its diverse manifestations arise from multiple pathways working together to produce the disease’s complex clinical patho-phenotype.

## 1. Introduction

Alzheimer’s disease (AD) is a clinical-pathophysiological entity, currently conceptualized as a continuum [1]. Pathologically, AD is characterized by cortical atrophy, neuronal loss, neuroinflammation, and synaptic degeneration [2]. Its two hallmark pathological features include the accumulation of neurofibrillary tangles (NFTs) and β-amyloid peptide deposits [3]. These molecular and cellular alterations are strongly linked to the manifestation of clinical symptoms like dementia [4]. AD is commonly categorized as early-onset (EOAD) or late-onset (LOAD) based on an age cut-off, typically 65 years [5]. LOAD is the most prevalent neurodegenerative disorder in older adults. It affects 5–10% of this population, with prevalence rising to 50% in individuals over 85 years old [6]. Aging is the most important risk factor for the development of AD [7]. The rise in the aging population, along with enhanced longevity, underscores the imperative for the prevention and treatment of progressive neurodegenerative diseases, particularly LOAD [8]. LOAD has been linked to disruptions in multiple biological pathways, often occurring simultaneously during disease progression. The condition presents significant complexity in both its underlying causes and clinical symptoms. The diverse molecular phenotypes observed in complex diseases such as LOAD arise from the coaction between genetic predispositions and environmental factors, which together influence various pathways and processes. For example, genetic variants associated with LOAD can impact multiple biological systems, including immune dysfunction, altered lipid metabolism, and neuroinflammation [9,10]. Additionally, a spectrum of other prevalent and complex disorders has been linked to an increased risk of developing Alzheimer’s disease [11,12,13,14,15], including other psychiatric disorders [16,17,18]. Research in this area has significantly advanced our understanding of the biological foundation of Alzheimer’s disease, particularly its association with β–amyloid and tau-related mechanisms. However, the precise molecular events and biological pathways driving the disease remain elusive. This genetic heterogeneity poses substantial challenges for the development of targeted therapies and the identification of disease-associated molecular changes. Moreover, the limited understanding of how these pathological molecular events interconnect further impede the discovery of effective therapeutic targets. Given the intricate genetic interactions involved in LOAD, the condition presents a compelling case for investigation through network analysis.

Post-mortem diagnosis of AD is staged by the severity and distribution of these pathological hallmarks: extracellular β-amyloid deposits and intracellular neurofibrillary tangles (NFTs) in neurons [19]. Tangles initially emerge in the entorhinal cortex (Braak stages I–II), subsequently spreading to the hippocampus and thalamus (Braak stages III–IV), and ultimately reaching the neocortex (Braak stages V–VI). This progression generally correlates with the worsening of cognitive function, ranging from mild cognitive impairment to severe dementia [19,20,21]. The dorsolateral prefrontal cortex (DLPFC) is located in the middle frontal gyrus (MFG) of the frontal lobe of the neocortex. It’s an area in the prefrontal cortex, which is not an anatomical structure, but a functional one, located in the lateral part of the 9 and 46 Brodmann areas (BA) [22]. DLPFC does not operate in isolation; it interacts with various brain regions, including the thalamus [23], hippocampus [24], and associative areas like the posterior temporal and parietal cortices and it is recognized for its role in working memory [25], speech [26], cognitive control [27], cognitive flexibility [28], planning [29], impulse inhibition and self-control, risk/benefit evaluation [30], problem solving [31], among other high order cognition functions. These functions are impaired during AD progression [32], and this region has been proposed as a therapeutic target to improve working memory in individuals with AD [33]. For these reasons, the DLPFC was selected as the focus of our analysis due to its role in cognitive processes most susceptible to impairment in AD and its potential as a target for therapeutic interventions.

### Transcriptional Networks

Networks are basic tools in systems biology for expressing the components of biological systems as whole integrated systems [34]. Many natural systems are structured in the form of networks. For example, genes do not function in isolation, but rather as part of large information processing networks that involve multiple molecular entities [35,36] involving dynamic events, which are subject to regulation and integration at multiple levels, each with multiple steps. In this way, gene expression is a complex process that requires the coordinated regulation of multiple steps, from DNA to RNA, including transcription initiation, elongation, termination, and post-transcriptional modifications. To better understand this complexity, gene co-expression networks offer a powerful framework to investigate biological systems, such as expression landscapes in disease [37].

Gene expression can be structured as a matrix thus characterizing gene expression data as multivariate. When converting multivariate data into a network, it becomes essential to apply metrics or some form of measurement capable of quantifying the distance or similarity between two models [38]. Using a correlation measure, we can examine how genes are co-expressed within an expression matrix, resulting in the construction of an adjacency matrix that is mathematically homomorphic to a co-expression network. Transcript co-expression networks often exhibit complex topological structures and a high degree of heterogeneity in their connection strengths and capacities. These networks are distinguished by the presence of “hubs”—nodes with a disproportionately large number of connections compared to the average degree and therefore have a robust architecture as they are resistant to random node failure [39]. However, the presence of hubs makes scale-free networks vulnerable to targeted insults where a hub is compromised [40]. In co-expression networks, hub genes frequently contribute to the stability and organization of the system. Furthermore, gene organization and expression are structured within modular structures [41], where certain genes within these communities of genes can serve as drivers of specific functionalities and organizational patterns. By analyzing relevant network metrics, the graph’s topological structure can be leveraged to gain insights into the macro, mesoscopic and local landscape of health and disease. These network relationships, when considered collectively, offer a snapshot of a phenotype’s state and provide a basis for comparing distinct phenotypes [42].

Transcriptome network analysis has been a tool in understanding large-scale dysregulations associated with complex diseases such as AD and dementia [43]. In addition, downstream network analysis has been useful in nominating potential therapeutic candidates and uncovering disease-specific patterns [44,45] and recognizing genes related to medical characteristics [46,47], even at the level of individual cells, revealing a set of astrocyte-related genes involved in choline metabolism and polyamine biosynthesis, which are associated with cognitive reserve and its influence on AD pathology [48].

We aim to identify key metrics that characterize the disease-associated network structure and organization, ultimately highlighting the biology (genes, transcripts and biological pathways) involved in the pathological processes of late-onset Alzheimer’s disease (LOAD), specifically in the dorsolateral prefrontal cortex (DLPFC). Building on these observations, this study presents a statistical analysis of a co-expression network model of the DLPFC associated with LOAD. By leveraging RNA-seq data and probabilistic modeling, we develop both functional and semi-mechanistic models of the disease.

## 2. Results

Table 1 presents demographic and genetic characteristics of two subject groups: control individuals (n = 307) and individuals diagnosed with pathological Alzheimer’s Disease (AD) (n = 486). Both groups show similar educational backgrounds but slightly differing mean ages. Sex distribution reveals a higher number of females in both groups, with a greater proportion in the pathological AD cohort. APOE genotype distribution shows a significant presence of the 33 genotype in both groups, with an increased occurrence of risk-associated genotypes (34 and 44) in the AD group. The study distribution consists of subjects from the ROS and MAP studies, with a relatively balanced representation across groups. Finally, racial composition is predominantly White, with a small representation of Black or African American individuals and minimal representation from other racial groups.

The graph models were constructed as indicated in the methodology and the degree distribution and global connectivity of both were analysed. Both graphs showed a *power-law* behavior, meaning that a small number of nodes, or hubs, are highly connected, while the majority have fewer connections (See Figure 1). Both networks are essentially disassortative. The Scaling exponent (γ) in the log-log plot indicates that the AD network has a slightly higher slope, implying that the AD network tends to have more nodes with a higher number of connections than the control network. This is confirmed in the cumulative frequency representation, where we see the same phenomenon but represented in a cumulative degree graph.

It is relevant to notice that the inferred co-expression networks are *undirected* as they were built by using mutual information which is a symmetric measure of statistical dependency. Hence, explicit mechanisms of regulation (such as inhibition or activation) are not directly disclosed from these networks. However, in some cases this information can be supplemented with additional omic information (perhaps from ChIP-seq or TF binding experimental data) to better reflect the full scope transcriptional regulation processes [49,50,51].

In the AD network, certain genes or nodes play a central role in maintaining connectivity across the system. In contrast, the control network shows a slightly more homogeneous and decentralized structure. This reflects a more balanced distribution of connectivity, where fewer nodes have a large number of connections, leading to a network that is less dependent on a small set of hubs for its overall cohesion. The K-S test showed that there is no statistically significant evidence to claim that the underlying distributions of the two samples are different (D = 0.037916, *p*-value = 0.9971). Table 2 shows the topological features of both networks.

Edges of the AD and control networks shared only 68.39% similarity. Also, two key centrality metrics, degree centrality and betweenness centrality, were analyzed to identify key genes in the topology of the networks. For both metrics, the 10% of genes with the highest degree and betweenness were taken. We found 112 Hub genes in the AD network and 108 in the control network. Both models share 82% of the Hub genes, but there are 11% of genes found only in the AD network. In parallel, 134 and 129 High betweenness centrality genes were found in the AD and control network, respectively. 52% of the high betweenness centrality genes are common to both conditions, while 25% of the genes were found only in the AD network (see Figure 2a,b). Only the high betweenness genes enriched pathways in the Biological Process (BP) ontology test. Pathways enriched included primarily the synaptic vesicle cycle, synaptic vesicle maturation, regulation of synaptic vesicle exocytosis and vesicle-mediated transport in synapse (see Figure 2e). Nine hub genes identified in the control network were no longer classified as hubs in Alzheimer’s disease. These genes were enriched in *peptide cross-linking*, *spliceosomal tri-snRNP complex assembly*, and *skin barrier establishment*.

### Mesoscopic Network Analysis

Graph partitions showed 68 modules of genes being coexpressed for the AD graph and 71 modules for the control graph (see Figure 3). By comparing the *Q* value for each network partition it is possible to evaluate whether there is a network in which the module detection algorithm, applied to each network, identifies a more modular structure [52]. The Q of the AD graph is slightly higher (0.28) compared to the control graph (0.20), suggesting that the AD network has a clearer community division. We further analyzed the modular structure of the networks by comparing the community partitions derived from the *Infomap* algorithm, which yielded a NMI of 0.4479.

A similarity matrix based on the membership of genes to their respective modules was constructed to understand the rearrangement of genes into different modules. Of the modules in both networks, 10 modules were preserved with perfect similarity between the two networks. In addition, 11 modules showed a similarity of more than 90%, 12 modules had a similarity greater than 80%, 14 modules exceeded 70%, 18 modules showed a similarity greater than 60%, and 24 modules achieved a similarity of more than 50%. This points to a reorganization of genes in several modules between the two networks. Although some of the modules remain largely intact, in others a significant redistribution of genes between different communities is observed (See Figure 4a).

Other similarity matrix, now that compares the enriched biological functions of individual modules between the two networks to evaluate the similarity of biological functions across the networks (Figure 4b). This compares the enriched biological functions of individual modules between the two networks. Gene Ontology (GO) terms for Biological Processes (BP) were assigned to each module, allowing for a detailed functional analysis. A similarity matrix was generated based on the GO terms associated with the enriched modules. Each entry in the matrix represents the degree of functional similarity between modules in the two networks. Modules with highly similar biological functions between the two networks were identified. 25 modules were found to be enriched with the same GO terms and there are 52 modules with at least 50% similarity in GO terms.

Furthermore, to ascertain the representation of each biological process within the network, we quantified the number of modules associated with a particular biological process based on GO enrichment analysis. We determined the quantity of modules representing each biological process, offering a perspective on the distribution of these processes throughout the network (Figure 5).

## 3. Discussion

### 3.1. Gene Co-Expression Network Alterations in Alzheimer’S Disease: Structural and Connectivity Insights

We used the ROSMAP cohort because it represents a broad-ranging spectrum of the older adult population compared to most autopsy series, which typically rely on selective recruitment criteria and individuals seeking medical care for their symptoms. This provides more representative and generalizable results. Additionally, we selected the DLPFC for this study due to its involvement in various higher cognitive functions, such as decision-making, working memory, and executive control, which are particularly relevant in AD, as impairments in these functions are hallmark features of the disease.

Previous transcriptomic analysis has revealed gene regulatory interactions occurring in brain tissues or regions of healthy and diseased individuals. For example, the impact of Alzheimer’s disease on different cortical regions has been investigated by gene co-expression network analysis and module and trait network analysis [45,46,53]. Here, two models of gene-to-gene correlations in the DLPFC of individuals with Alzheimer’s disease and a control group composed of older individuals were analyzed to examine alterations in coexpression patterns associated with Alzheimer’s pathology. Networks were constructed following restrictions that made them comparable. Both networks maintain the same number of connections and have a very similar size and diameter.

The K-S test indicated no significant statistical difference between the global degree distributions of the two transcriptional profiles. This suggests that the overall degree distribution remains largely unchanged. However, while the degree distributions appear statistically similar, differences may still emerge at mesoscopic and micro-level network properties. It is important to consider that both distributions correspond to the same anatomical regions in aged individuals, which may influence their structural and functional characteristics. Nonetheless, it is noteworthy that the AD network exhibits a higher number of highly co-expressed genes (see Figure 1). In contrast, the control network shows a more balanced distribution of connectivity where fewer nodes have a high degree of connections, indicative of a network that is less dependent on a small set of hubs for its overall cohesion.

Only around two-thirds of the gene-gene interactions are shared between the AD and the control network. Gene interaction patterns overlap substantially, but there is still about 31.61% difference in overall connectivity. However, this difference is key in differentiating between the normal and AD-related transcriptional regulation landscapes. The shared edges may represent preserved connections essential for basic brain function or transcriptional network neighborhoods less affected by AD pathology.

Additionally, the control network exhibits a slightly higher transitivity compared to the AD network (See Table 2). While this difference is modest, the reduction in transitivity in the AD network may be showing a lower degree of local cohesion in the gene coexpression relative to the control network.

### 3.2. Functional Insights into Co-Expression Changes in the AD Network: Linking Epigenetic, Cytoskeletal, Immune, and Post-Transcriptional Pathways

In the context of co-expression networks, node degree can be interpreted as an indicator of a gene’s influence within the regulatory program. In the AD network, 13 hub genes were identified that are absent in the control network. These genes, listed in order of degree, include KRT6B, COL6A5, PDCD5P1, H3C9P, FCRL1, ONECUT3, CCL18, H2BC14, H2AC4, CROCC2, SLC25A24P1, PDE4DIP and FCRL3 (See Appendix A for further details). A brief description of these genes and their relations with AD is listed in Appendix A. Among these, KRT6B, located on chromosome 12, stands out as the most highly connected gene in the AD network, with a degree of 207.

From the 13 hub genes here, KRT6B, COL6A5 and CROCC2 are structural proteins and cytoskeletal components; H3C9P, H2BC14, H2AC4 and ONECUT3 are histones or genes related to chromatin regulation; FCRL1, FCRL3 and CCL18 are related to the immune system and inflammation; PDCD5P1 and PDE4DIP are pseudogenes related to neurodegeneration and cell death and SLC25A24P1 is a pseudogene of SLC25A24, involved in ATP transport and energy metabolism.

Interestingly, we can find 4 pseudogenes in the list of AD-specific-hub genes (SLC25A24P1, PDE4DIP, SLC25A24P1 and H3C9P). Studies have demonstrated that pseudogenes may participate in gene transcription post-transcriptional gene regulation and epigenetic regulation [54]. Other computationally predicted pseudogenes have been proposed to be involved in the etiology of AD and other NDs [55] and other pseudogenes have been identified as hubs in AD networks [45]. These findings suggest that pseudogenes may act as regulatory elements. Investigating the specific roles of these pseudogenes could provide deeper insights into their contributions to AD pathophysiology and highlight novel therapeutic targets for future research.

Other interesting gene is CCL18. It is known that chemokines and their receptors can promote AD pathology by inducing the production of Aβ and some chemokines and receptors are also involved in tau phosphorylation. CCL18, a well described chemokine was found to have enhanced co-expression with other genes in the AD network. Given its enhanced co-expression with other genes in the AD network, CCL18 might act as a mediator or regulator of neuroinflammatory responses. For example, the chemokine-receptor interaction could stimulate pathways that promote the production or Aβ accumulation [56].

The nine hub genes identified in the control network, which were no longer classified as hubs in Alzheimer’s disease, were enriched in three key biological processes: peptide cross-linking, spliceosomal tri-snRNP complex assembly, and skin barrier establishment. Peptide cross-linking refers to the covalent bonding of two functional groups within a single protein (intramolecular) or between two proteins (intermolecular), often stabilizing protein structures or facilitating interactions between domains [57]. Spliceosomal tri-snRNP complex assembly involves the aggregation, arrangement, and bonding of one or more small nuclear RNAs (snRNAs) with multiple protein components to form a ribonucleoprotein complex. This complex plays a crucial role in the formation of the spliceosome, which is essential for precursor mRNA (pre-mRNA) splicing and gene expression regulation [58].

We also investigated the biological processes associated with the first neighbors of the hubs specific to the AD network. On average, AD-specific hub genes co-express with 30.15 new genes. Among these, CROCC exhibited the greatest increase in co-expression within the AD network, showing interactions with 66 newly co-expressed genes (See Figure 6a).In contrast, PDE4DIP displayed the fewest new co-expression connections, with only three additional genes. We identified genes that were exclusively co-expressed with AD-specific hubs in the AD network. We found a set of newly co-expressed genes that were consistently gained across all AD-specific hubs. Notably, 83 genes were found to be commonly co-expressed with hubs in the AD network. Functional enrichment analysis of these newly associated neighbors revealed four main functional groups (See Figure 6b). The first group consists of processes related to *nucleosome organization and assembly*, primarily involving several histone genes. A key mechanism of epigenetic regulation in gene expression is the dynamic organization of chromatin structure, and epigenetic aberrations have been implicated in transcriptional alterations associated with AD [59].

The second group, which can be categorized as *cytoskeletal organization and keratinization*, is functionally related to the first group and includes genes from the KRT family as well as FLG. There is evidence suggesting that KRT family genes may play a role in neurological disorders, including dementia [60]. Research has demonstrated that the dysregulated expression of Keratin 9 is a consequence of AD pathology. However, its extensive interaction network implies additional, as yet unidentified downstream effects that may contribute to the initiation and progression of the disease [61]. Furthermore, keratin intermediate filaments have multiple roles in signaling pathways, inflammation, and various disease states [62]. Interestingly, the high-degree gene KRT6B has been implicated in the regulation of epithelial–mesenchymal transition (EMT) and immune-related pathways, including macrophage differentiation and polarization, as demonstrated in studies of bladder cancer [63].

This leads us to the third functional group, which includes antimicrobial and humoral immune response. In the past, research on inflammation-driven neurodegeneration and the immune role of Aβ has given rise to the “Antimicrobial Protection Hypothesis” of AD. This hypothesis suggests that β-amyloid deposition represents an early innate immune response to actual or misinterpreted immunological threats. Initially, Aβ captures and neutralizes invading pathogens by forming β-amyloid deposits. The subsequent fibrillization of A Aβ activates neuroinflammatory pathways, aiding in pathogen clearance and β-amyloid removal. However, in AD, prolonged activation of this response results in persistent inflammation and neurodegeneration [64].

Finally, the fourth group is related to miRNA-mediated post-transcriptional gene silencing, a key pathway in the regulation of mRNA stability and translation, involved in fine gene control and adaptive response processes. The first three groups of metabolic pathways are mediated by at least one gene, and give rise to an explanation of the interpolation between at least two hypotheses of Alzheimer’s neuropathogenesis, including the antimicrobial protection hypothesis [59,65] and the neuroinflammation hypothesis [66]. The antimicrobial humoral response and the keratinization pathways of genes are connected by SPINK5, a serine protease inhibitor [67] and KRT6A, one of the 27 different type II keratins expressed in humans [68], while the immune response and nucleosome assembly pathways are at least connected by the H2BC10 histone.

### 3.3. High Betweenness Genes Only Present in the Disease Network Are Involved in Diverse Biological Pathways

In the AD network, 44 genes high betweenness centrality genes were identified that were not central in the control network (See Appendix A). Ten genes with higher betweenness were LOC124903002, RAB3A, SNAP91, BEX1, INPP5F, NECAP1, INA, ATP5F1B, NSG2 and SYP. LOC124903002, an uncharacterized gene, was the gene with higher betweenness. The gene appears in 5454.812 shortest paths between pairs of other nodes. Pathway enrichment analysis revealed that these AD-specific high-betweenness genes are primarily associated with processes related to synaptic function. Specifically, enriched pathways include the regulation of *short-term neuronal synaptic plasticity*, the *synaptic vesicle cycle*, *synaptic vesicle maturation*, the *regulation of synaptic vesicle exocytosis* and the *establishment of vesicle localization*. Beyond these, 20 other pathways were identified, all related to vesicle dynamics, exocytosis, transport, and regulatory mechanisms. It’s notable that high-betweenness centrality genes are enriched in synaptic processes, as synaptic processes are essential for sustaining the flow of information within the brain metabolism. These genes, acting as “bridges” or nodes of elevated information flow within the network, are playing a structurally analogous role to that of synaptic functions, which similarly manage and modulate information transmission across neural circuits.

In contrast, there are 39 genes that are serving as high information-passing nodes in the control network but are not shown to be highly mediated genes in the pathology network. Two nodes, KIFBP and ACTR2 are not even present in the AD network, due to their poor statistical dependency with other genes. These “missing” genes did not enrich any gene ontology as a group. The 10 higher betweenness centrality genes that are lost in the AD network as high degree-brokering genes include MAPK9, SYT3, PACSIN1, GNG3, FBXL19, AMPH, SERPINI1, RUSC2, AP2M1, and KIFBP. The Mapk9 mitogen-activated protein kinase 9 is the most highly mediated gene lost in the AD network. This gene has functions in the transduction of environmental signals to the transcriptional machinery, a function similar to those of other MAP kinases, and it phosphorylates the c-Jun protein [69], an essential component of the AP-1 transcription factor, highlights its involvement in stress-activated pathways, such as apoptosis, inflammation, and neuroprotection. The loss of MAPK9 (betweenness in control 5714.584, betweenness in AD network 738.7555) as a high betweenness gene in the AD network is particularly relevant because it implies a disruption in a key signaling pathway involved in cellular stress responses and transcriptional regulation, probably having extensive implications of network reconfiguration in AD and the potential functional vulnerabilities that may emerge from the disruption of signaling intermediates.

One of these genes, SERPINI1 (betweenness of 2179.33 and 570.30 in control and AD, respectively) encodes neuroserpin, an inhibitory serine protease crucial for regulating plasmin and other processes related to neuronal homeostasis. The aggregation of mutant neuroserpin polymers—resulting from the misfolding of neuroserpin—in the brain is associated with the formation of abnormal intracellular inclusions, leading to neuronal degeneration and dementia [70,71]. It’s required for normal synaptic plasticity [72] and has demonstrated neuroprotective properties, as it reduces infarct volume and shields neurons from ischemia-induced apoptosis [73]. Neuroprotective effects may result from its role in inhibiting excitotoxicity, reducing inflammation, and preventing blood-brain barrier disruption after acute ischemic stroke [74]. The node corresponding to SERPINI1 experienced a reduction of 1609 in the number of shortest paths that pass through it, maybe reflecting a loss of neuroprotective capacity and contributing to the neuronal degeneration characteristic of AD.

Another gene that lost its status as a high-intermediate node was SYT3. Synaptotagmins are a class of Ca^2+^-dependent synaptic vesicle membrane proteins prevalent in the brain, potentially involved in membrane trafficking [75]. SYT3 is present in presynaptic terminals. It is expressed throughout the brain and is enriched in the brainstem, cerebellum and hippocampus [76]. Ischemic conditions have been reported to upregulate SYT3 expression [77]. Given the link between AD and impaired cerebral circulation—characterized by reduced oxygen and nutrient supply— Chum et al. [78] proposed that cerebrovascular miRNAs associated with AD regulate mRNA targets crucial for endothelial cell functions such as angiogenesis, vascular permeability, and blood flow regulation. Their findings revealed that several miRNAs associated with AD also regulate both cardiovascular and neuronal pathways, with SYT3 identified as a gene targeted by more than four miRNAs. This overlap shows a probable relationship between vascular and neuronal health in AD. Genes such PACSIN1, AMPH, RUSC2 and KIFBP participate in mechanisms associated with cytoskeleton dynamics and intracellular transport [79,80,81,82]. The absence of these genes as high-betweenness nodes indicates a disturbance in their function as mediators within the network, potentially hindering the coordination of cytoskeleton remodeling and intracellular trafficking pathways. Such abnormalities may lead to deficiencies in neuronal architecture, synapse functionality, and transport mechanisms, which are essential processes in neurodegenerative disorders such as AD.

### 3.4. Network Functional Analysis Show Gene Modular Rearrangement

The idea that sets of genes carrying out related tasks ought to be strongly correlated is supported by the clustering of genes in highly associated modules [83]. It has been seen that in gene co-expression models, different gene modules may correspond to distinct biological processes [84], where functionally related genes tend to co-cluster within specific modules.

The control network exhibited a greater number of modules compared to the AD network. However, the higher modularity strength of the AD network suggests that the communities of coexpressed genes in the AD network are more clearly separated from each other, even though they are fewer in number (See Table 2). This suggests that interactions between gene co-expression modules, which indicate the biological compartmentalization of the transcriptomic program, become more isolated in disease conditions, with weaker internal connectivity within modules.

As a result, cross-talk between different gene communities is reduced, perhaps indicating a breakdown in the coordinated control of genes. This pattern is further supported by differences in transitivity. The control network shows a higher local clustering, indicating that local gene neighborhoods are more tightly interconnected. The combination of higher modularity and lower local clustering in the AD network suggests a more fragmented structure: while gene communities are more isolated from each other, the internal connectivity within modules is weaker. In contrast, the control network displays modules that are not only internally cohesive but also well-integrated with other modules. This shift in network architecture in AD could reflect disrupted regulatory coordination and a loss of efficient information flow across gene communities possibly leading to breakdowns in communication.

NMI indicates a 45% of similarity between the two graphs partitions. The similarity matrix based on the correspondence of shared genes between these modules revealed a reorganization of genes across various modules. While the modules in the AD and control networks are not directly equivalent, there is a degree of similarity between certain modules in both networks, as we can see in the diagonal of the matrix. However, a marked difference in the arrangement of gene membership across modules is apparent (see Figure 4a). In parallel, the biological similarity matrix for the communities displays a more conserved pattern across modules. Although some modules show distinct biological functions, indicating possible divergence in their roles across different biological contexts, the functional organization remains relatively preserved despite the substantial reorganization of genes across modules (see Figure 4b).

Differences in the transcriptional regulatory landscape between AD and healthy aged individuals are more subtle than in other diseases such as cancer, where we can see a clearer disruption in the modularity and coexpression [85]. The overall differences in topology point to changes at the mesoscopic level in the rearrangement of genes and molecular functions in co-expression modules in AD.

Upon quantifying the representation of biological functions within the modules, we found that most biological processes exhibit significantly higher representation in the AD network compared to the control (See Figure 5).

Processes that expanded in the AD network were related to neuron formation, axon or dendrite development, and synaptic activity included *Axonogenesis, Axon development, Neuron projection guidance, Axon guidance, Dendrite development, Regulation of synapse structure or activity, Regulation of synapse organization, Neuron projection extension, Neuron migration, Dendritic spine development, Central nervous system neuron differentiation, Cerebellum development*. Since significant axonal damage occurs at amyloid plaques, secondary spine loss likely results from presynaptic dysfunction. In AD, synaptic dysfunction is a primary driver of cognitive decline and memory deficits [86,87].

Additionally, processes related to neurodegeneration and apoptosis included *Neuron apoptotic process, Regulation of neuron apoptotic process, Response to amyloid-beta, hindbrain development, metencephalon development*. Cognitive functions such as *Cognition, Learning or memory* and metabolic and transport-related functions, such as the *Nucleobase-containing compound catabolic process* were also observed to expand.

Nevertheless, although fewer in number and of lesser magnitude, some processes exhibit the opposite trend. BPs that reduced their representation in pathological condition included those related to small particles like *Amide transport, Amino acid transmembrane transport, Organic anion transport*. These processes involve proteins such as ion channels, transporters (uniporters, symporters, antiporters), and pumps, which facilitate the movement of these molecules across biological membranes [88]. Additionally, other secretion- and transport-related biological processes, including *Regulation of protein secretion, Regulation of protein secretion, Regulation of secretion by cell, Regulation of peptide transport and secretion, Regulation of monoatomic ion transport, Regulation of hormone secretion, Organic anion transport* also exhibited expansion.

The processes with gene modularity spreading associated with the pathology primarily pertain to neuronal development, neurodegeneration, axon guidance, and cognitive function. In contrast, the transcriptomic profile of control samples is more closely related to metabolic regulation, cellular transport, secretion control, and extracellular responses.

Axonogenesis and axon development display the greatest divergence in module representation, being distributed across 10 modules in the control network and 19 modules in the AD network. This response may be linked to the loss of the SERPINI1 (Neuroserpin) gene as a high-betweenness gene. As mentioned before, neuroserpin is recognized to be important for axonal development and neurite outgrowth [72].

In AD, cells, especially neurons and glial cells, undergo significant changes in their environment due to factors such as the accumulation of beta-amyloid plaques, NFTs and neuroinflammation [89]. These changes generate a chronic cellular stress environment, which can trigger adaptive responses in gene expression. To cope with this adverse environment, the transcriptional program could be reorganized, promoting the activation and redistribution of genes co-expresssion. This phenomenon can also reflect a possible compensatory or adaptive strategy to neurodegeneration.

## 4. Methods

### 4.1. Data Acquisition and Classification

A complete workflow for this study is depicted in Figure 7. RNA-seq count matrices from the dorsolateral prefrontal cortex (DLPFC), specifically encompassing Brodmann area BM46 and part of BM9, were obtained from 793 subjects (473 with AD and 301 controls) through the Religious Orders Study and Memory and Aging Project (ROSMAP) database [90]. These datasets were downloaded from the AMP-AD Knowledge Portal (https://www.synapse.org) in compliance with all applicable terms and conditions.

RNA sequencing was performed following the methodology described in De Jager et al. [91]. Libraries were prepared using rRNA depletion, and sequencing was conducted on an Illumina NovaSeq 6000 platform. Metadata was used to classify patients into Control and AD groups according to the AD neuropathologic changes, following the NIA-AA criteria as outlined in the revised guidelines for the neuropathologic assessment of brain disorders common in the elderly, including Braak staging for assessing the severity of neurofibrillary tangles and the Consortium to Establish a Registry for Alzheimer Disease (CERAD) semi-quantitative measure for neuritic plaques (CERAD score) [92].

### 4.2. Quality Control

Quality control of the data was performed using the NOIseq package in R [93]. After a diagnostic analysis, transcript composition bias, and length bias were corrected. Genes with low counts were filtered using a threshold of 1 CPM. Batch effects were corrected using the ASCA Removal of Systematic Noise on Seq data (ARSyNSeq) method, with the ’sequencingBatch’ column as the reference. ARSyNSeq filters the noise associated with identified or unidentified batch effects considering the experimental design and applying Principal Component Analysis (PCA) to the ANOVA parameters and residuals [94]. After this processing 28,263 features remained, including protein-coding genes, long non-coding RNAs, microRNAs, pseudogenes, and other RNA species. The entire dataset was then normalized by the trimmed mean of the M-values normalization method (TMM), which also adjusts by transcript length [95] (See Appendix A). Finally, data was stratified by diagnosis, resulting in two expression matrices.

### 4.3. Inference of Co-Expression Networks

Mutual Information-based gene coexpression networks were constructed separately for two groups: individuals diagnosed with pathological AD and age-matched controls. Nodes in the network represent genes, and edges represent significant coexpression relationships between them based on Mutual Information (MI). MI is the maximum entropy/maximum likelihood estimate of statistical dependence between two random variables [96]. MI has attractive information-theoretic interpretations and can be used to measure non-linear associations in complex systems [97], such as transcriptional programs. This characteristic is advantageous since many relationships between transcripts do not follow a linear statistical dependence. As MI is well-defined for discrete or categorical variables, we first discretized the counts and posteriorly calculated the mutual information by gene pairs with the Infotheo R package [98]. To establish a MI heuristic cut, basic network metrics by percentile were obtained for percentiles 99.9999%, 99.999%, 99.99%, 99.9%, 99%, 98%, 90%, and 80% (See Appendix A). Also, we ran a null model to certify that the network had characteristics not due to randomness (See Appendix A)

By analyzing various network metrics across different percentiles (e.g., number of vertices, edges, components, transitivity) provides a clear picture of how the structure changes as we move from lower to higher MI interactions. The 99.99% percentile retains a manageable number of nodes and edges while maintaining the largest components and preserving important topological features while filtering out less meaningful interactions. By selecting interactions at the 99.99% percentile, we ensure that only the statistic dependence (i.e., those with the highest MI values) are included in the network. This helps to filter out noise and weaker associations that may not contribute meaningfully to the biological interpretation. This approach is equivalent to stratifying the data based on *p*-value thresholds [99]. The networks were then analyzed using the igraph package in R version 2.0.3 [100] and Cytoscape v3.10.2 [101].

### 4.4. Topological Analysis and Network Centralities Measure

A common approach for analyzing specific network properties is to compute and compare certain local or global network indices such as degree distribution and connectedness [102]. We began by identifying key quantities that characterize the structure and organization of the networks. First, we evaluated the degree distribution behavior in both models. The degree of a node represents the number of direct connections it has with other nodes. We further compared the cumulative degree distributions between the two models to assess topological differences. A Kolmogorov Smirnov (K-S) test was performed to test whether both distributions were significantly different. We performed a goodness-of-fit test to determine the best-fitting distribution for our data. Various distributions, including power-law, Poisson, normal, Pareto, exponential, and negative binomial, were evaluated based on statistical criteria (See Appendix A).

To compare the overall connectivity differences between the models, the Jaccard similarity for the edges was calculated by comparing the sets of edges in each network. The computed index shows the ratio of the intersection of edges between the two networks to their union, as described in Equation (Equation 1).(1)$$J\left(E_{g_{1}},E_{g_{2}}\right)=\frac{|E_{g_{1}}\cap E_{g_{2}}|}{|E_{g_{1}}\cup E_{g_{2}}|}$$Equation  (Equation 1): The Jaccard Index equation was used to compare global connectivity between both networks.

Genes with high connectivity were identified based on node degree and betweenness centrality in the models. In coexpression networks, *Hub genes* are genes with a significantly high number of direct connections to other genes, serving as hyper-connected nodes that contribute to network stability and functional coordination. We defined hub genes as those with the highest node degree, specifically the top 10% of the degree distribution. Moreover, betweenness centrality measures how often a node appears on the shortest path between pairs of other nodes, indicating its importance in controlling information flow across the network [40]. By occupying a central position in the network, they influence the flow of biological information, signals, and resources, ensuring efficient communication between different regions. Consequently, they can either facilitate interactions or act as bottlenecks within the system. Accordingly, high betweenness centrality genes were defined as those ranking in the top 10% of betweenness values. To explore the biological functions of the AD-specific hub and high betweenness genes, we performed a GO enrichment analysis using the *clusterProfiler* package in R. We focused on Biological Process (BP) ontology terms with a *p*-value cutoff of 0.05 and a q-value cutoff of 0.05.

Biological enrichment analysis (often called over-representation analysis, ORA) usually works (as is the case with *clusterProfiler*) by performing hypergeometric tests to estimate the extent and likelihood that one set (in our case the list of selected genes) is over-represented in another set (e.g., the list of genes annotated in a given biological GO-category or Pathway). Likelihoods are calculated as drawn from a hypergeometric distribution. *p*-values are calculated for the null model and represent the probability of accepting the null hypothesis of random occurrence. q-values correspond to the false discovery rate (FDR) correction for multiple testing of these *p*-values.

To gain a deeper understanding of connectivity changes related to hubs, we identified and enriched the first neighbors of the hubs specific to the AD network that were also exclusively first neighbors of hubs within the same network.

To visualize the connectivity of AD-specific hub genes, we extracted an induced subgraph comprising only the hub nodes from the AD network. This subgraph highlights the dense interactions between these key genes (see Figure 2c). In parallel, an induced subgraph comprising only the high betweenness nodes from the AD network was built (See Figure 2d).

Transitivity (an approximation to the global clustering coefficient) [102] was computed for both control and problem networks. This is a measure used to detect the fraction of triplets that are closed in directed networks [103]. In other words, transitivity measures the probability that the adjacent vertices of a vertex are connected [104], measuring how tightly connected a node’s neighbors are.

### 4.5. Inference of Modular Structure and Functional Analysis

Modularization seeks to address complexity by decomposing a problem into smaller, conceptually manageable units. In efforts to understand the organization and function of complex systems such as transcriptional programs, researchers often identify communities or modules within biological networks, simplifying them into conceptually distinct entities representing compartmentalized biological processes. This approach facilitates the study of phenotypes by deconstructing these systems into manageable components. To identify modular substructures in both models, networks were partitioned into gene modules using the *Infomap* algorithm [105,106]. The Infomap algorithm is a community detection method for complex networks based on information theory. It optimizes a map equation, which minimizes the description length of a random walk on the network, effectively partitioning nodes into communities where information flow is more contained. The algorithm simulates a random walker navigating the network and detects communities where the walker tends to stay longer. The main idea is that good community structures allow efficient compression of the walk’s trajectory, minimizing this description length. It can detect sub-communities within larger clusters, something relevant for biological function discovery. It is efficient for large networks and outperforms modularity-based methods (e.g., Louvain) in certain cases [107,108]. It has been previously used to infer communities [109,110], and compare networks in biological contexts to other algorithms [111].

To assess the strength of modular divisions within each network, we computed the global Newman modularity (Q) for both networks using the modularity function from the igraph package. Q measures the quality of a given partition by quantifying how well-defined the communities are—i.e., how distinct and segregated different groups of nodes are from one another [112]. Q measures the strength of community division rather than just counting the number of communities. It ranges from −1 to 1, with 1 indicating optimal modularity. A high Q value means that nodes within the same community are densely interconnected, while connections between different communities are sparse [113]. We also computed Q per module in both networks, to understand whether the modularity differences arise from local connection reorganization like hub genes migration rather other topological changes (See Appendix A).

We used normalized mutual information (NMI) as a community detection comparison metric to assess differences in the modular structure of AD and control networks. NMI was computed using the compare() function in the igraph package. Then, to evaluate the similarity of genes and biological functions across the entire network based on module enrichment, we constructed two similarity matrices: one representing gene correspondence to modules and another capturing enriched modules according to their associated biological functions. Functional similarity was quantified using the Jaccard index of semantic similarity scores, allowing for an objective comparison of the biological processes represented within each module by their respective functions.

To determine the representation of each biological process across the network, we quantified how many modules (or communities) are associated with a specific biological process following Gene Ontology (GO) enrichment analysis. We identified the number of modules in which each biological process is represented, providing insight into the distribution of these processes across the network (See Figure 5).

## 5. Conclusions

### Linking Basic and Clinical Research

In recent times a shift has risen in the way research in AD is contextualized. In particular with reference to the way basic research guides clinical trials, for instance. Crucial insights into the clinical relevance of disease-modifying therapies (DMTs) for AD, emphasizing the importance of robust outcome measures and patient-centered interpretations of clinical trial results has been developed by initiatives such as the European Union-North American Clinical Trials in Alzheimer’s Disease (EU-US CTAD) Task Force [114]. This perspective is particularly relevant for a study on our work in differential transcriptional programs and modular network rearrangements in late-onset AD, as it underscores the necessity of linking molecular alterations to clinically meaningful phenotypes. The discussion on phase III anti-amyloid trials highlights the importance of longitudinal assessments and network-level changes, which aligns with the study’s focus on modular network reorganization in AD progression. Moreover, the emphasis on biomarker-informed evaluations reinforces the need to integrate transcriptional and network-based findings with therapeutic response markers, ultimately contributing to a deeper understanding of how molecular disruptions translate into functional decline.

Moreover, because our correlations are based on the statistical dependence of mutual information, we can detect nonlinear relationships. However, these interactions lack directional inference, meaning we cannot determine whether the correlation is indicates activation or inhibition. Additionally, AD is a disease with high comorbidity, a factor not explicitly accounted for in our analysis. Integrating additional regulatory information -such as ChIP-seq or TF binding data or ingle cell omics- may significantly enhance mechanistic interpretation. Also, other models such as Agent-Based Models (ABMs) could be used to simulate the interactions of individual agents to assess their effects on the system as a whole [115].

To develop a more comprehensive understanding of the complex transcriptional mechanisms underlying AD, further exploration of brain regions—particularly those most affected by proteinopathy—is necessary. Our findings should further be viewed as in-silico genomic-data-driven hypotheses that require additional experimental validation, given the probabilistic nature of the differential connectivity relationships.

## Figures and Tables

**Figure 1 ijms-26-02361-f001:**
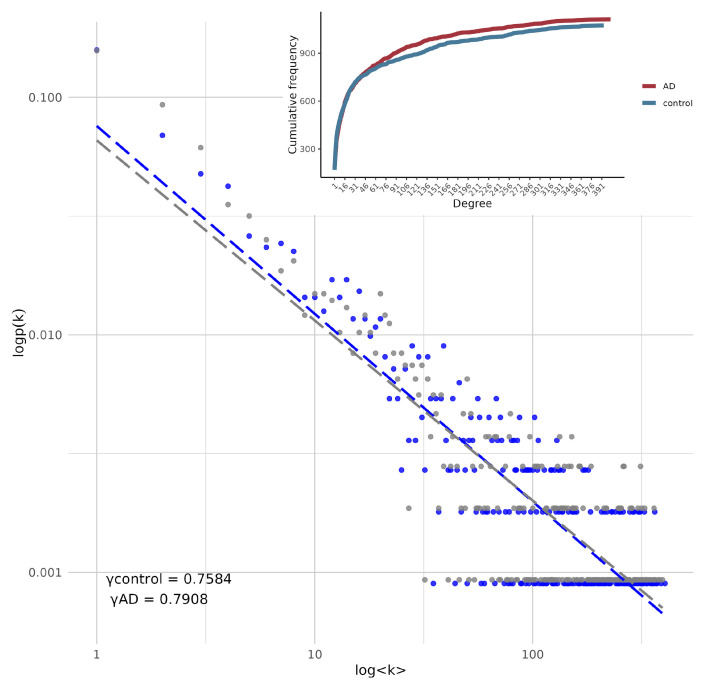
log(k) vs log(p(k)). log-log representation of the degree distribution for the two networks. Lines and dots in blue represent the data of the network corresponding to AD and nodes in gray represent the data of the control network. (Inset) Cumulative degree distribution of both models. The red curve represents the AD network and the blue curve the control network.

**Figure 2 ijms-26-02361-f002:**
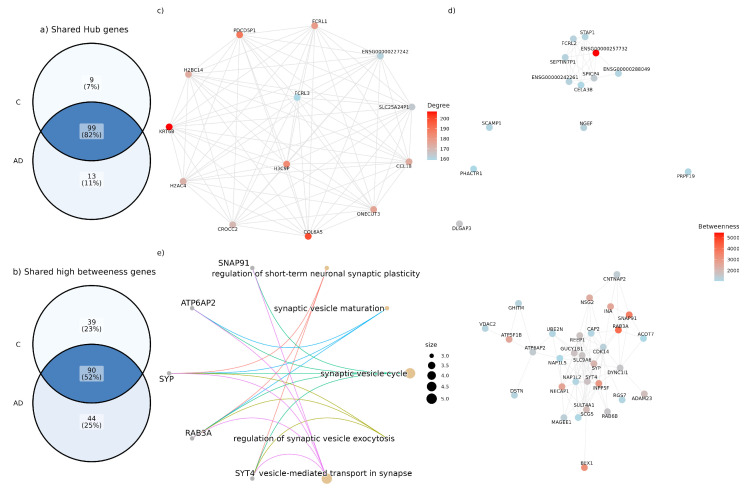
(**a**) Comparison of shared and exclusive “hub” genes between people with AD (AD) and controls (C). The blue area shows the hub genes shared by both conditions, while the other sectors highlight the exclusive genes. (**b**) Comparison of genes with high betweenness centrality between people with and without AD. The blue intersection indicates genes with high shared betweenness centrality, while the remaining sectors show genes unique to each condition. (**c**) Induced subgraph of hub genes present in the network of people with AD that are not found in the list of hub genes of people without AD. This represents the genes specific to the AD condition in terms of degree centrality. (**d**) Induced subgraph of genes with high betweenness centrality present in the network of people with AD, which are not in the list of “hub genes” of people without AD. (**e**) Gene-Concept Network of the enriched pathways for the set of high betweenness genes in the hypergeometric test for the Biological Process (BP). Enriched pathways include the regulation of short-term neuronal synaptic plasticity, synaptic vesicle maturation, synaptic vesicle cycle, regulation of synaptic vesicle exocytosis, and the vesicle-mediated transport in synapse, among others.

**Figure 3 ijms-26-02361-f003:**
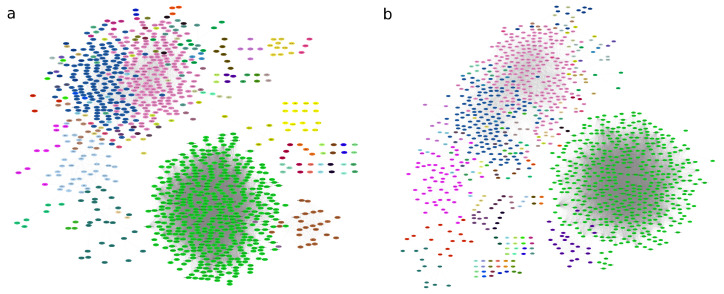
Modular structure of networks for (**a**) Control network; (**b**) Alzheimer’s disease coexpression network. The nodes belonging to a community are colored the same. The same color for different modules is not related.

**Figure 4 ijms-26-02361-f004:**
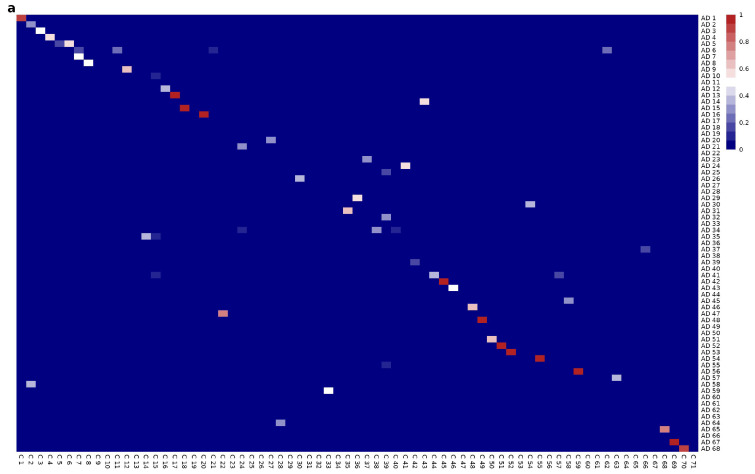
(**a**) Similarity matrix between gene modules, based on the correspondence of shared genes between these modules. Each row and column corresponds to a specific gene module from two different networks. The colors of the cells represent the degree of similarity between the modules, calculated using the Jaccard index. (**b**) Heatmap of functional enrichment similarities between gene modules based on GO:BP biological processes.

**Figure 5 ijms-26-02361-f005:**
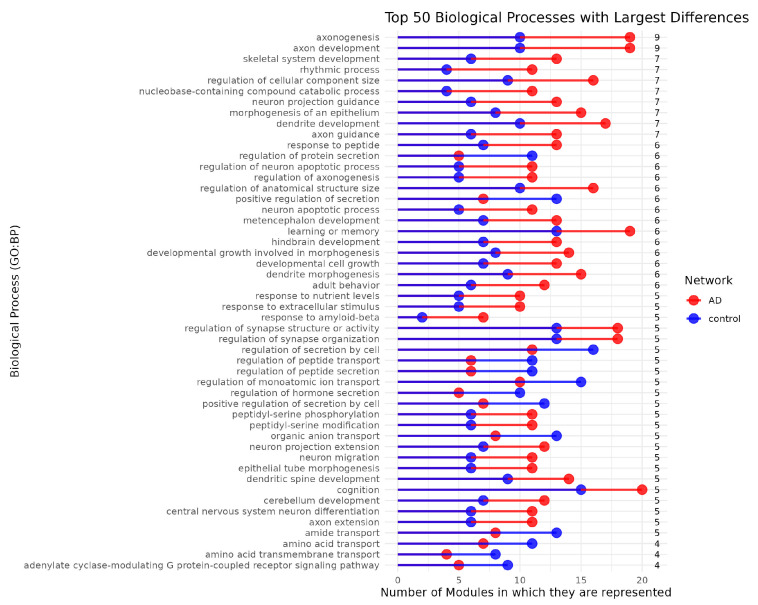
**Top 50 Biological Processes with Largest Differences in Module Representation between AD and Control Networks**. Here is shown the number of modules associated with the top 50 GO:BP terms that exhibit the largest differences in module representation between AD and control networks. Biological processes are listed on the *y*-axis, while the number of modules associated with each process is represented on the *x*-axis. Red bars correspond to modules in the AD network, while blue bars represent modules in the control network. The numbers on the right indicate the absolute differences of modular representation units.

**Figure 6 ijms-26-02361-f006:**
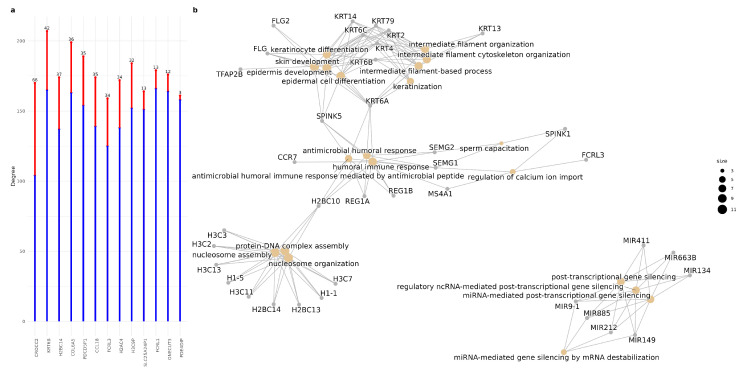
(**a**) Changes in the degree of hub genes in the control network (blue) and in the AD network (red). The number at the top of the bars indicates the difference in degree between the two conditions (**b**) Enrichment of common newly coexpressed genes. We can observe at least 4 weekly groups of metabolic pathways.

**Figure 7 ijms-26-02361-f007:**
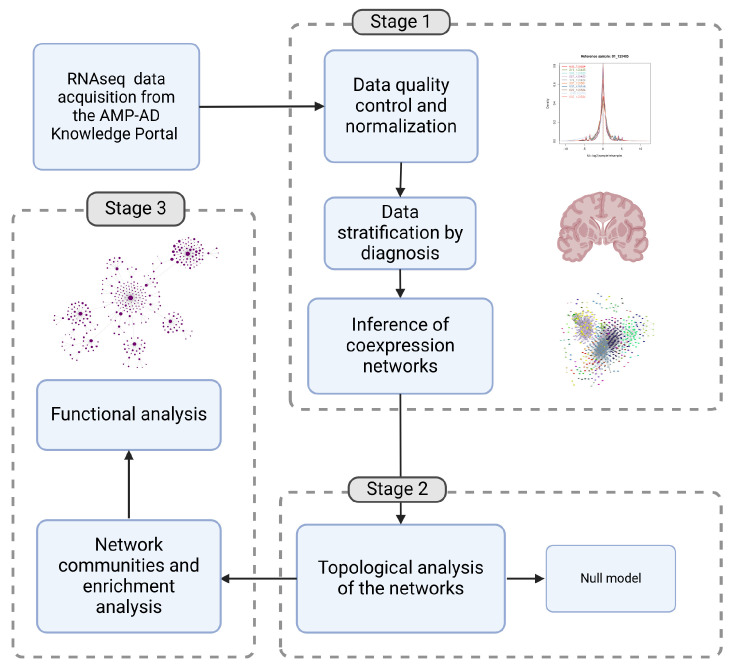
The workflow methodology used in this work includes 3 stages. **Stage 1** includes Data quality control, normalization and stratification, followed by the construction of the gene co-expression networks separately for the two groups: individuals diagnosed with AD and age-matched controls. **Stage 2** includes the Topological analysis and the null model test and finally, **Stage 3** includes communities and functional analysis (Created in BioRender.com).

**Table 1 ijms-26-02361-t001:** Demographic and genetic characteristics of control and pathological AD subjects.

	Control Subjects, n = 307	Pathological AD Subjects, n = 486
**Age**	85.8 ± 5.20	87.8 ± 3.67
**Years of education**	16.5 ± 3.59	16.1 ± 3.61
**Sex**		
Males	125	140
Females	182	346
**APOE genotype**		
22	3	0
23	61	54
24	4	11
33	209	274
34	28	129
44	0	12
Unknown	2	6

**Table 2 ijms-26-02361-t002:** Topological features of the AD and control network.

	Control Network	AD Network
Total genes	1074	1113
Number of edges	28,160	28,160
Network diameter	12	13
Global transitivity	0.6252799	0.5956005
Edges similitude (by Jaccard index)	68.39%	
Number of genes in largest connected component	529	568
Number of modules (Infomap partition)	71	68
Scaling exponent γ	0.7584	0.7908
Number of modules	65	71
Modularity (Q)	0.2027528	0.2834913

## Data Availability

ROSMAP data can be requested at http://www.radc.rush.edu. All code involved in this work is publicly available at https://github.com/CSB-IG/ROSMAP_MultiOmics/tree/main/ROSMAP-rnaseq-networks (accessed on 3 March 2025).

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
