# Peer review of "Differential Transcriptional Programs Reveal Modular Network Rearrangements Associated with Late-Onset Alzheimer’s Disease"

_ijms, 2025, doi:10.3390/ijms26052361_

Round 1
Reviewer 1 Report
Comments and Suggestions for Authors
See attachment.

NA
Author Response
Reviewer 1
The authors are grateful to Reviewer 1 for the insightful comments made about our work. We will follow the advice and counsel given. In what follows we will present a point-by-point response (our answers appear in bold-type to ease reading) to the review.
This study explores the transcriptomic network reorganization in late-onset Alzheimer's disease (LOAD) through gene co-expression network analysis. The topic is clinically relevant, and the methodology is relatively systematic. However, the following issues need further clarification or improvement to meet the publication standards of the journal:
- Data Representativeness and Confounding Factors Control
The sample description does not explicitly clarify the age matching details between AD patients and the control group (e.g., mean and standard deviation). Furthermore, were factors such as gender, education level, or other comorbidities (e.g., cardiovascular diseases) included as covariates in the analysis? These factors could influence transcriptomic data, and the methods for controlling these variables should be specified.
Thank you for your comment. We have added a table presenting the demographic characteristics included in both probabilistic models. We do not incorporate variables such as sex or educational level, as this would have required the construction of phenotype-specific networks, which in turn would have significantly reduced the sample size. This reduction is not recommended for the construction of a robust probabilistic network model. In addition, we were not provided with metadata on comorbidities, which we recognise as a limitation of our approach. We appreciate your suggestion and acknowledge the potential impact these factors may have on transcriptomic data. We have revised our manuscript to clarify these issues.
- Contradictory Interpretation of Network Topological Differences
The AD network exhibits higher modularity (Q value), but there is no significant difference in the global degree distribution (K-S test). The study should explore whether the modularity differences arise from local connection reorganization (e.g., increased hub genes within specific modules) rather than global topological changes.
We have made some additional calculations to explore this relevant question. The following information has been added as Section 5.2 of the Supplementary Materials:
In brief, modularity (Q) in a network, relative to a specific partition, quantifies the extent to which different vertex types are distinct or segregated from one another. To explore whether the global modularity differences arise from local connection reorganization, we calculated Q per module for both networks. We then examined the relationship between Q and the number of hubs within each module. This helps to compare the structural organization at the module level .
Larger modules exhibit higher modularity due to their greater number of connections, allowing for better distribution. In both of our models, hub genes—defined as the top 10\% of genes with the highest co-expression—are found exclusively in the largest module (Module 1), which is also the most modular. While modularity tends to decline as module size decreases, this relationship is not monotonic.
The overall modularity differences between the two networks primarily arise from differences in the largest modules. The modularity of the largest module in the control network was 0.17, compared to 0.23 in the AD network, indicating greater compartmentalization. Larger modules not only have higher modularity but also contain more hubs, as a higher number of nodes increases the likelihood of some becoming hubs while also providing more non-hub nodes that can connect to them. These modularity differences align with the observation that the AD network has more hubs, as seen in the degree distribution. In summary, it is possible that the differences in global modularity in the two networks are not due to the migration of hubs to smaller modules, but probably to other genes involved in global connectivity.
We have added a section to the supplementary materials to explore this observation. We have also added a better description of the methodology for computing Q in the methods section.
- Insufficient Functional Validation of Hub Genes
Pseudogenes (such as SLC25A24P1, H3C9P) are listed as AD-specific, but functional validation of these genes remains insufficient.
The sequencing libraries for this RNA-seq experiment were made using rRNA depletion, which results in transcript counts of other, lesser-known RNA species such as pseudogenes. The N for the construction of the networks in both cases together with the selection of nodes with the strongest statistical dependence should be sufficient to be confident that the nodes presented as hubs have probabilistic validity. We chose a limit of 10% of the most connected nodes as the definition of hubs, and these pseudogenes passed all the filters mentioned above. In addition, in the Supplementary materials we have summarised what is previously known about these pseudogenes and their relationship to AD. In the specific case of SLC25A24P1 and H3C9P, there is currently no additional literature linking these pseudogenes to AD pathology. However, their association in the proposed model networks suggests a potential avenue for future research.
- Mutual Information Cannot Infer Regulatory Direction (e.g., Activation/Inhibition)
It should be clarified that the results only reflect correlation, as mutual information cannot infer regulatory direction. The study should also discuss how the integration of regulatory networks (e.g., ChIP-seq or TF binding data) could enhance the mechanistic interpretation.
Thank you for your comment. We are aware of the directional constraints associated with mutual information, and we have included a note in the manuscript. Furthermore, we have provided additional clarification on how incorporating other regulatory interactions may enhance the mechanistic interpretation of the disease.
- Literature Citation Updates and Novelty
Some key methods (such as the Infomap algorithm) are referenced from older sources (2009), and recent comparative studies of community detection algorithms (e.g., Leiden algorithm) should be added. Additionally, the latest advances in AD research (such as cell-type specific networks revealed by single-cell transcriptomics) are not cited; it is recommended to include them to support the discussion.
Thank you for your suggestion. The reference by Emmons et al., 2013 (doi: 10.1103/PhysRevE.100.022301), along with the 2009 reference was cited as a more recent source for the Infomap algorithm. We also mentioned the Mathys et al., 2024 (doi: 10.1038/s41586-024-07606-7) as a work where networks have been analysed at the single cell level.
Most of the research in our lab involves computational inference and analysis of co-expression networks. We have performed numerous benchmarks of modularity detection methods, some of these tests are reported in a thematic review paper we published recently (doi: 10.3389/fgene.2021.701331). In these benchmarks, the Infomap and Leiden/Louvain algorithms are consistently the most efficient for biological module detection and their performance is comparable. It is, however, relevant to reference these works.
- Language and Logical
Some expressions are redundant (e.g., the repeated emphasis on "modular reorganization" in the discussion), and it is suggested to streamline the writing. Furthermore, terminology needs to be standardized (e.g., is "transitivity" being used interchangeably with "clustering coefficient"? This should be verified in the methodology section).
We have carefully revised our writing to correct these issues. The "transitivity" and "clustering coefficient" use of terms was homogenized, and discussion has been revised to avoid redundancy. We have also proceeded with a professional copyediting of the whole manuscript.
Reviewer 2 Report
Comments and Suggestions for Authors
The authors should provide some details about the computational power applied for the results achieved( GPU, High Performance Computing).

Author Response
Reviewer 2
The authors want to acknowledge Reviewer 2 for the professional academic reviewing made about our work. We will follow the comments and suggestions given. In what follows, we will present a point-by-point response (our answers appear in bold-type to ease reading) to the review.
Comments and Suggestions for Authors
The authors should provide some details about the computational power applied for the results achieved ( GPU, High Performance Computing).
The inference of the adjacency matrices was performed using a multicore setup with 40 threads, utilizing the future_map() function from the 'furrr' R package (v0.3.1). The total computation time was 13.2 hours, and no GPU was used. This was run on an 80 CPU Linux 5.15.0-126-generic server.
The article investigates how Alzheimer's disease (AD) is influenced by complex interactions between genetic and environmental factors. Using network-based approaches, the authors analyzed the transcriptional structure in the brain, identifying disease-specific markers and significant differences between AD and control networks. The results reveal a molecular reorganization that could help understand the causal mechanisms of AD, suggesting that its diverse manifestations arise from multiple interconnected biological pathways.
My comments are appended below:
1) 2.3. Inference Of Co-Expression Networks
The author discusses Node and Edge organization. Do they apply the GNN (graph Neural Network) architecture ?
No, although GNNs algorithms are often used in similar problems, we have decided to perform an information-theoretically based probabilistic modeling approach. We present the reasons as follows:
GNNs provide a structured way to learn from graph-structured biological data, where nodes represent genes and edges denote inferred regulatory or co-expression relationships. Their advantages for GCN inference include:
Traditional co-expression methods (e.g., Pearson correlation) capture only linear relationships, while GNNs can model complex, non-linear gene interactions using deep learning architectures.
GNNs can integrate multimodal datasets, including RNA-seq, epigenomic, and proteomic data, leading to more robust gene network inference. They can be trained on one dataset and generalized to other conditions or tissues.
Unlike static co-expression methods, GNNs use message-passing techniques to incorporate neighborhood information, improving the representation of gene relationships within biological pathways. GNNs can be combined with autoencoders or attention mechanisms to refine inferred networks by denoising or focusing on high-confidence interactions. However, GNNs require extensive training data, and their interpretability remains a challenge compared to traditional statistical approaches.
Probabilistic methods, particularly MI-based techniques, provide a complementary approach for gene co-expression inference. MI is a non-parametric measure that quantifies statistical dependency between gene expression profiles without assuming linearity. Unlike correlation-based methods, as much like GNNs, MI detects both linear and nonlinear dependencies between genes, making it effective for inferring complex gene regulatory interactions. MI-based approaches are less sensitive to noise compared to deep learning methods, especially when applied to biological data with heterogeneous variance.
The accuracy of MI estimation improves (with theoretical guarantees) provided, as sample size increases, reducing bias and variance in network inference. Methods such as Minimum Redundancy Maximum Relevance (mRMR) and ARACNE (Algorithm for the Reconstruction of Accurate Cellular Networks) refine MI-based inference by filtering indirect associations.
MI values can be easily interpreted in the context of information theory, unlike deep learning models, which require post hoc explainability techniques (e.g., SHAP values in GNNs). MI estimation is computationally expensive, especially for high-dimensional datasets. Requires sufficient sample sizes to ensure reliable MI calculations; otherwise, estimation noise may introduce spurious edges in the inferred network. Since our sample sizes and computational capabilities allowed it, we decided to use an MI based approach.
In particular Graph Neural Networks (GNNs) are effective for analyzing network-based data and understanding the interactions between genes and pathways.
Moreover I would suggest the use of the AlphaFold framework could be very useful! AlphaFold, developed by DeepMind, is an artificial intelligence program that predicts protein structures with remarkable accuracy. By using AlphaFold, researchers can model the interactions between proteins and other molecules, which could provide further insights into the molecular mechanisms of Alzheimer's and help identify new therapeutic targets(https://en.wikipedia.org/wiki/AlphaFold, https://deepmind.google/technologies/alphafold/alphafold-server/).
AlphaFold uses the Transformer model, which includes attention mechanisms. In particular, AlphaFold exploits a combination of graph networks and attention mechanisms to analyze the relationships between amino acids in proteins.
Thank you for this suggestion. While the approach of using protein-nucleic acid interactions is extremely useful to unveil a certain class of gene regulatory networks - technically, those based on transcription factor binding site (TFBS) predictions- where molecular binding is key for gene expression. However, the class of gene networks we are more interested in for this work is probabilistic gene co-expression networks. The reason is that genes may co-express for several reasons. Sharing molecular binding motifs (for which the suggested AlphaFold approach would be optimal) is one of these. However, “functional” gene co-expression in which genes are co-expressed obeying signaling and metabolic constraints conditioned by a cascade of underlying biological processes contingent on the phenotypes (e.g. Alzheimer-type disease versus control). In such cases, and provided a large enough sample size is given (which is true for the data from the ROSMAP collaboration we used), using conditional probabilistic modeling results more advantageous.
2) 2.4. Topological analysis
Which are the implication of p-value cutoff of 0.05 and a q-value cutoff of 0.05.? The authors mention various distributions, including power-law, Poisson, normal, Pareto, exponential, and negative binomial, were evaluated. They can show the importance of other distributions like Poisson or exponential ones.
Biological enrichment analysis (often called over-representation analysis, ORA) usually works by performing hypergeometric tests to estimate the extent and likelihood that one set (in our case the list of selected genes) is over-represented in another set (e.g., the list of genes annotated in a given biological GO-category or a Pathway). Likelihoods are calculated as drawn from a hypergeometric distribution (i.e. a multivariate set of Fisher exact tests). p-values are calculated for the null model and represent the probability of accepting the null hypothesis of random occurrence. q-values correspond to the false discovery rate (FDR) correction for multiple testing of these p-values. We have added a couple of paragraphs clarifying this in the revised manuscript.
3) 2.5. Inference of Modular Structure
The authors can describe the InfoMap in detail . Did the authors apply the Entropy function/ theory ?
The Infomap algorithm is a community detection method for complex networks based on information theory. It optimizes a map equation, which minimizes the description length of a random walk on the network, effectively partitioning nodes into communities where information flow is more contained. The algorithm simulates a random walker navigating the network and detects communities where the walker tends to stay longer. The main idea is that good community structures allow efficient compression of the walk’s trajectory. The way it works is that the map equation quantifies the minimum description length required to encode a random walk on the network. The algorithm finds the optimal partitioning that minimizes this description length. Infomap naturally identifies multi-level community structures, i.e., it can detect sub-communities within larger clusters, something relevant for biological function discovery. It is efficient for large networks and outperforms modularity-based methods (e.g., Louvain) in certain cases.
In our lab, we have performed numerous benchmarks of modularity detection methods, some of these tests are reported in a thematic review paper we published recently (doi: 10.3389/fgene.2021.701331). In these benchmarks, the Infomap and Leiden/Louvain algorithms are consistently the most efficient for biological module detection and their performance is comparable.
4) Figure 5 can be enhanced the character on x,y axis.
Figure 5 has been improved to enhance readability.
5) Figure 6 some of the 50 Biological processes can be described in detail.
Following the reviewers’ advice, key biological processes from figure 6 have been thoroughly discussed in the revised manuscript.
6) The author can cite this reference: https://www.sciencedirect.com/science/article/pii/S2274580724006514
Thank you for pointing out this relevant reference that has been revised and cited in the discussion as follows.
In recent times a shift has risen in the way research in AD is contextualized. In particular with reference to the way basic research guides clinical trials, for instance. Crucial insights into the clinical relevance of disease-modifying therapies (DMTs) for Alzheimer's disease (AD), emphasizing the importance of robust outcome measures and patient-centered interpretations of clinical trial results has been developed by initiatives such as the European Union-North American Clinical Trials in Alzheimer's Disease (EU-US CTAD) Task Force. This perspective is particularly relevant for a study on our work in differential transcriptional programs and modular network rearrangements in late-onset AD, as it underscores the necessity of linking molecular alterations to clinically meaningful phenotypes. The discussion on phase III anti-amyloid trials highlights the importance of longitudinal assessments and network-level changes, which aligns with the study's focus on modular network reorganization in AD progression. Moreover, the emphasis on biomarker-informed evaluations reinforces the need to integrate transcriptional and network-based findings with therapeutic response markers, ultimately contributing to a deeper understanding of how molecular disruptions translate into functional decline.
7) The authors can consider some control models and theories that could be valuable to consider in their study on Alzheimer's disease:
Agent-Based Models (ABMs): ABMs simulate the interactions of individual agents to assess their effects on the system as a whole. They are particularly useful for systems that are not well understood and can model complex biological interactions, https://direct.mit.edu/posc/article/29/4/468/107062/Biological-Control-Variously-Materialized-Modeling
Thank you for pointing out this reference and the methods discussed therein, we have included these ideas in the revised discussion.
Round 2
Reviewer 1 Report
Comments and Suggestions for Authors
The authors have addressed the major concerns raised during the previous review round through substantive revisions.
Reviewer 2 Report
Comments and Suggestions for Authors
The manuscript can be accepted for pubblication